Insights on Pinna nobilis population genetic structure in the Aegean and Ionian Sea

Sarafidou Georgia g.sarafidou@hcmr.gr 1 2
Tsaparis Dimitris 2
Issaris Yiannis 1
Chatzigeorgiou Giorgos 2
Grigoriou Panos 3
Chatzinikolaou Eva 2
Pavloudi Christina 4 5
1 Institute of Oceanography (IO), Hellenic Centre for Marine Research (HCMR) , Anavyssos , Greece
2 Institute of Marine Biology, Biotechnology and Aquaculture (IMBBC), Hellenic Centre for Marine Research (HCMR) , Heraklion , Crete , Greece
3 Cretaquarium, Hellenic Centre for Marine Research (HCMR) , Heraklion , Crete , Greece
4 PSL Research University: EPHE-UPVD-CNRS, UAR CNRS 3278 Centre de Recherche Insulaire et Observatoire de l’Environnement (CRIOBE) , Perpignan , France
5 Laboratoire d’Excellence “CORAIL”, Centre de Recherche Insulaire et Observatoire de l’Environnement (CRIOBE) , Moorea , French Polynesia
Ward Eric
Electronic publication date: 2023 Nov 29
Publication date: 2023
Volume: 11
Electronic Location ID: e16491
Received 2023 Jan 24; Accepted 2023 Oct 29
Copyright: ©2023 Sarafidou et al.
Copyright year: 2023
Copyright holder: Sarafidou et al.
License: This is an open access article distributed under the terms of the Creative Commons Attribution License, which permits unrestricted use, distribution, reproduction and adaptation in any medium and for any purpose provided that it is properly attributed. For attribution, the original author(s), title, publication source (PeerJ) and either DOI or URL of the article must be cited.
License URL: https://creativecommons.org/licenses/by/4.0/

Keywords: Pinna nobilis, Population genetics, Mediterranean, COI, 16S rRNA, eDNA, Critically endangered, Haplotype networks, Genetic structure

Funding: Transnational Cooperation Programme Interreg V-B “Balkan-Mediterranean 2014-2020” MIS 5017160 Natural Environment and Innovative Environmental Actions 2020 This work was supported by the RECONNECT project [MIS 5017160] financed by the Transnational Cooperation Programme Interreg V-B “Balkan-Mediterranean 2014–2020” and co-funded by the European Union and National Funds of the participating countries and by the PINNA STATUS project, funded by the Hellenic Green Fund and its funding program “Natural Environment and Innovative Environmental Actions 2020”, in the Priority Axis “Biodiversity Conservation Actions”. The funders had no role in study design, data collection and analysis, decision to publish, or preparation of the manuscript.

==============================
The fan mussel Pinna nobilis Linnaeus, 1758 is an endemic species of the Mediterranean Sea, protected by international agreements. It is one of the largest bivalves in the world, playing an important role in the benthic communities; yet it has been recently characterized as Critically Endangered by the IUCN, due to mass mortality events. In this context, the assessment of the genetic variation of the remaining P. nobilis populations and the evaluation of connectivity among them are crucial elements for the conservation of the species. For this purpose, samples were collected from six regions of the Eastern Mediterranean Sea; the Islands of Karpathos, Lesvos and Crete; the Chalkidiki and Attica Peninsulas; and the Amvrakikos Gulf. Sampling was performed either by collecting tissue from the individuals or by using a non-invasive method, i.e., by scraping the inside of their shells aiming to collect their mucus and thus avoid stress induction to them. Conventional molecular techniques with the use of the COI and 16S rRNA mitochondrial markers were selected for the depiction of the intra-population genetic variability. The analyses included 104 samples from the present study and publicly available sequences of individuals across the whole Mediterranean Sea. The results of this work (a) suggest the use of eDNA as an efficient sampling method for protected bivalves and (b) shed light to the genetic structure of P. nobilis population in the Eastern Mediterranean; this latter knowledge might prove to be fundamental for the species conservation and hence the ecosystem resilience. The haplotype analyses reinforced the evidence that there is a certain degree of connectivity among the distinct regions of the Mediterranean; yet there is evidence of population distinction within the basin, namely between the Western and the Eastern basins. The combination of both genetic markers in the same analysis along with the inclusion of a large number of individuals produced more robust results, revealing a group of haplotypes being present only in the Eastern Mediterranean and providing insights for the species’ most suitable conservation management.

Introduction

During the autumn of 2016 a massive mortality phenomenon was observed on the Western Mediterranean populations of Pinna nobilis, the largest endemic bivalve of the Mediterranean Sea (Darriba, 2017). The mass mortality events (MME) reached quickly the Eastern Mediterranean Sea (Katsanevakis et al., 2019). Although several pathogens have been proposed as the MME agents (Catanese et al., 2018; Carella et al., 2019; Carella et al., 2023; Panarese et al., 2019), the most likely one is the protozoan Haplosporidium pinnae, which is considered to affect the digestive gland of the animal, resulting in stress, starvation and in a general dysfunction and finally death of the organism (Box, Sureda & Deudero, 2009; Grau et al., 2022). Based on all the above, the species status in the IUCN red list changed to Critically Endangered (CR) (Kersting et al., 2019).

The decline of the pen shell’s population was known several years before the MME (Centoducati et al., 2007) due to threats such as the coastal construction activity, the degradation of its habitats, the anchoring—especially at touristic hotspots, the wave action, the byssus exploitation for production of sea silk and the illegal trawling activity (Hendriks et al., 2013; Basso et al., 2015). Therefore, a series of regulations were established aiming to protect this species and ensure its survival; national legislation and international conventions have been in force for the past decades, such as the Barcelona Convention for the Protection of the Marine Environment and the Coastal Region of the Mediterranean and the Council Directive 92/43/EEC on the Conservation of natural habitats and of wild fauna and flora (Annex IV). Nevertheless, the effectiveness of those measures was argued, since P. nobilis was still subject to illegal fishing for personal or commercial consumption or for decorative purposes (Katsanevakis et al., 2011).

Undoubtedly, P. nobilis is a beneficial species for the benthic communities for a number of reasons, since it offers various ecosystem services. As a filter feeder, it filters large amounts of water contributing to seawater clarity (Basso et al., 2015), a process that benefits the meadows of the cohabitant species Posidonia oceanica and/or Cymodocea nodosa (Trigos et al., 2014). Its large valves provide a hard substrate within a sandy area for many sedentary organisms, so it is fairly considered as an ecosystem engineer (Rabaoui et al., 2015). It sometimes also cohabits with the crustaceans Pontonia pinnophylax or Nepinnotheres pinnotheres (Hassine, Zouari & Rabaoui, 2008; Akyol & Ulaş, 2015), thus increasing even more the complexity and species richness of the community in which it lives. Recently, due to the attention it has attracted, P. nobilis has been characterized as a flagship species (Scarpa et al., 2020). Without a doubt, this recognition is significant not only for the conservation of the species itself and the ecosystem it is associated with, but also for raising public awareness about marine environmental issues in general (Polgar & Jaafar, 2018).

P. nobilis has been the focus of numerous molecular studies conducted in various regions of the Mediterranean Sea over the past decades. A study by Katsares et al. (2008) revealed low genetic differentiation among the examined populations in Thermaikos Gulf (Greece), possibly attributed to the species’ pelagic larval stage and the resulting high gene flow. Similar findings were observed in studies conducted along the Tunisian coasts (Rabaoui et al., 2011), which also indicated the absence of a genetic barrier between the Aegean Sea and the Tunisian coasts. A study across a wider area of the Western Mediterranean by Sanna et al. (2013) provided additional insights on P. nobilis populations; it was the first one to include a considerable number of samples and, actually indicated a distinct genetic structure between the Western Mediterranean (Sardinia, Corsica, Sicily) and the Eastern Mediterranean (Aegean Sea and Tunisian coasts). Furthermore, it identified the Venetian Lagoon population from the northern Adriatic Sea as a potentially diverging population. Interestingly, two other areas in the central Adriatic Sea, the natural marine parks of Mljet and Telascica, showed greater similarity to the Western Mediterranean samples than to those from Venice (Ankon, 2017).

In 2015, microsatellite markers were used for the first time for P. nobilis samples from the Balearic coasts (González-Wangüemert et al., 2015) suggesting their usefulness for the genetic diversity and connectivity assessments. Wesselmann et al. (2018) combined both mitochondrial and microsatellite markers along with lagrangian simulations to suggest a series of insightful conclusions for the populational genetics of P. nobilis with the upper aim of enhancing its conservation. In the Gulf of Lion (North-Western Mediterranean Sea) P. nobilis populations exhibited high genetic diversity across various locations, although there was no significant genetic differentiation among these populations, thus indicating a genetically homogeneous population spanning the entire coastline (Peyran et al., 2021). Clearly, the small geographic scale surveys seem to indicate populational genetic homogeneity; however, on a larger Mediterranean scale, where a greater number of samples are included, the distinction becomes more evident.

The majority of these publicly available sequences are partial sequences of the mitochondrial DNA, and for the most part COI and 16S rRNA genes. Even though the mtDNA is more widely used for phylogeographic purposes, it can reveal a significant level of differentiation among and within populations, as has been shown in several studies for marine bivalves (Parker et al., 1998; Matsumoto, 2003; Wood et al., 2007; Feng et al., 2011; Fernández-Pérez et al., 2018; Ramadhaniaty, Setyobudiandi & Madduppa, 2018). It should also be noted, that mtDNA in certain bivalves, such as Donax trunculus (Theologidis et al., 2008) and Mytilus spp. (Zouros, 2013), has a biparental inheritance which, undoubtedly, affects population diversity estimates based on it.

The aim of the present study was to (a) investigate the genetic diversity of the P. nobilis populations in an area of the Eastern Mediterranean Sea that has not been investigated to date, and (b) compare it with similar studies from the whole Mediterranean in an attempt to provide further insights into population structuring of this critically endangered species, which will offer a good estimation on the fitness and diversity of the Greek populations.

Materials and Methods

Sampling area

For the purpose of the study 105 samples were collected within the period of August 2018–April 2021 from six locations of the Eastern Mediterranean Sea and particularly from the Islands of Karpathos, Lesvos and Crete, Vourvourou (Chalkidiki peninsula), Attica Peninsula, and the Amvrakikos Gulf (Fig. 1, Table S1). Depth at each collection point was recorded by the divers using a diving computer. Samples from Crete, Chalkidiki and Attica Peninsulas and Amvrakikos Gulf were collected under a relevant research permit (175828/2195 of 14/11/2018) issued by the Greek Ministry of Environment and Energy, General Directorate for the Forests & Forest’s Environment, Department of Wildlife and Hunting Management. Samples from Karpathos were collected under research permit 171978/1203 of 18/07/2018 issued by the Greek Ministry of Environment and Energy, General Directorate for the Forests & Forest’s Environment, Department of Wildlife and Hunting Management.

D. Karagiannis of the National Reference Laboratory for Mollusc Diseases (Greek Government) provided the samples MYT1-MYT9 from Lesvos, which were collected under a permit from local authorities (MEE/GDDDP89926/1117). Samples TS1–TS6 from Lesvos were collected under a permit by the Department of Agriculture and Fisheries, Decentralized Administration of the Aegean (No. 52321/6-9-2018).

Figure 1 Map of the sampling locations of the current study (numbers in square brackets indicate the number of samples).

Credits: Giorgos Chatzigeorgiou. CC0. Map created using the Free and Open Source QGIS.

eDNA sampling

The sampling method for Karpathos’ samples was non–lethal, non-invasive and low impact aiming at the minimization of the disturbance towards the bivalves, since the tissue removal may provoke stress and make the animal more susceptible to diseases. Initially, a rod of 0.5 cm diameter was placed at the opening of the valves of each animal by the SCUBA divers in order to keep them slightly open, carefully taking into account the fragility of the shell’s outermost part. Consequently, a sampling brush, resembling a buccal swab was used (Fig. S1) to scrape tissue remnants and mucus from the interior of the valves. The sampling brushes (one for each individual) were placed in small zip bags and stored at −20 °C until further processing. Additionally, the shells’ width and height (above and below sediment) were recorded by the divers using a caliper.

Tissue sampling

All the other samples were collected from sacrificed individuals under research permits, since the initial aim of the sampling was the investigation of the infection of P. nobilis from the parasite H. pinnae. Specifically, 50–100 mg of different tissues (mantle, gills, digestive gland) from each individual were removed, preserved in absolute ethanol and stored at 4 °C until further processing. As previously, the shells’ width and height were recorded by the divers, for the majority of the individuals.

DNA extraction

DNA was extracted according to the protocol of Sambrook, Fritsch & Maniatis (1989), and as previously described in Grau et al. (2022), both from the brushes as well as from the tissues. Specifically, in the case of the latter, small pieces of the collected tissues were chopped with sterile scissors; triplicate extractions were performed for each tissue. Each replicate sample was washed with 800 µl of sterile distilled water for 15 min, following centrifugation at 13,000 g for 2 min, as in Darriba (2017). The supernatant was removed and the wash was repeated. Afterwards, each sample was washed with 600 µl of lysis buffer (0.5 M Tris, 0.1 M EDTA, 2% SDS, ph 8.8) for 15 min, following centrifugation at 13,000 g for 2 min and removal of the supernatant. The washes with the lysis buffer were repeated twice. The pellet was mixed with 600 µl of lysis buffer and 6 µl of proteinase K (20 mg/ml) and incubated at 55 °C overnight. DNA was extracted by precipitation with isopropanol and ammonium acetate (5 M) (Sambrook, Fritsch & Maniatis, 1989). In the final step of the DNA extraction protocol, i.e., the elution of the DNA pellet, replicate samples were pooled and their concentration was measured in a NanoDrop 1,000 spectrophotometer (Grau et al., 2022); DNA concentrations are provided in Table S1.

PCR amplifications

For the PCR amplification of the tissue samples, no specific tissue was chosen but rather a mixture of all the extracted DNAs, for each individual, in similar concentrations. Initially PCR amplifications were performed for the COI and 16S rRNA genes with previously used primers and conditions (Folmer et al., 1994; Sanna et al., 2013; Sanna et al., 2014; Leray et al., 2013); however, the amplifications were not successful. Therefore, new primers were designed (Table 1) based on the available P. nobilis sequences in GenBank (Sayers et al., 2023).

Table 1 Primers used in the present study.

Target gene	Forward primer (5′–3′)	Reverse primer (5′–3′)	Amplification length (bp)	Reference	
COI	5′- CAGCTTTTGTAGAGGGCG - 3′	5′- CCAAATTACACCAGTCAGCC - 3′	722	this study	
5′- GATCCGGGATAGTAGGTAC - 3′	5′- CMGGATGACCAAARAACC - 3′	645	this study	
5′- ATGGCYGTCGATTTAGC - 3′	5′- CMGGATGACCAAARAACC - 3′	298	this study	
COI	LCO 1490	HCO 2198	710	Folmer et al. (1994)	
mlCOIintF	jgHCO2198	313	Leray et al. (2013)	
5′- GGTTGAACTATHTATCCNCC - 3′	5′- GAAATCATYCCAAAAGC - 3′	338	Sanna et al. (2013)	
16S rRNA	5′- GGTAGCGAAATTCCTAGCC - 3′	5′- AAKGGTSGAACAGACCC - 3′	408	this study	
16S rRNA	5′- TGCTCAATGCCCAAGGGGTAAAT - 3′	5′- AACTCAGATCACGTAGGG - 3′	450	Sanna et al. (2013)	
nad3	5′- CCTTATGARTGYGGBTTT - 3′	5′- TCHATAAGYTCATARTAYARCCC - 3′	203	Sanna et al. (2014)	

Each PCR contained 2 µl of DNA template (about 20 ng/ul), 4 µl of 5X KAPA HiFi Fidelity Buffer (Roche Molecular Systems, Inc., Basel, Switzerland), 1 µl of each primer (10 um), 0.8 µl of dNTPs (10 mM each), 1 µl of KAPA HiFi HotStart DNA Polymerase (1 U/uL) (Roche Molecular Systems, Inc., Basel, Switzerland) in a total volume of 20 ul. Amplifications were performed at a BioRad T100 thermal cycler. The PCR protocol was the same for the two genes; namely a denaturation step at 95 °C for 5 min followed by 35 cycles of 98 °C for 20 s, 53 °C for 30 s, 72 °C for 30 s and a final extension step at 72 °C for 5 min.

Amplification of the 16S rRNA yielded in some cases a double PCR product; in this case, purification of both the PCR products was carried out from a 2% agarose gel using the NucleoSpin Gel and PCR Clean-up (MACHEREY-NAGEL, Allentown, PA, USA). For the COI amplicons, a sodium acetate-absolute ethanol cleanup protocol was conducted. All purified PCR products were sequenced in an automated sequencer ABI 3730.

Analyses

The ABI chromatograms were checked and corrected by eye using the BioEdit Sequencing Alignment Editor software (Hall, 2011) and MEGA X sequence analysis software (Kumar et al., 2018). 16S rRNA sequences, COI sequences and concatenated 16S rRNA-COI sequences (following the approach of Sanna et al., 2013) from the present study were aligned with the Clustal W package (Thompson, Higgins & Gibson, 1994) embedded in BioEdit and MEGA X. In addition, publicly available sequences of the corresponding genes (COI and 16S rRNA) of P. nobilis, for which sample location information was available, were also downloaded from GenBank and added to the aforementioned alignments (Table S2; Fig. S2). However, it should be highlighted that each survey—where the sequences derived from—aimed at a distinct gene region, and this resulted in a small overlap when the sequences were aligned all together. Furthermore, some surveys did not include both of the genes amplified in the present study, i.e., they focused either only on the 16S rRNA gene or on the COI gene (Fig. S2). For these reasons, it was not possible to create only one dataset that could include all this information and therefore three datasets were created instead (Table 2).

Table 2 Genetic diversity estimates.

	Dataset	N	Bp	h	Hd	Ps	Pi	
1	16S rRNA-COI Greece (Eastern Mediterranean)	100	982	34	0.91 ± 0.017	45	0.00304 ± 0.00029	
2	16S rRNA-COI Central, Western and Eastern Mediterranean Sea	294	714	104	0.961 ± 0.005	72	0.00511 ± 0.00019	
3	COI Mediterranean Sea	450	243	48	0.652 ± 0.024	36	0.00475 ± 0.00028	
Notes.

N number of sequences

Bp Base pairs

h number of haplotypes

Hd Haplotype diversity

Ps Polymorphic sites

Pi Nucleotide diversity

Phylogenetic trees were constructed using the IQ-TREE web server (Trifinopoulos et al., 2016) with automatic identification of substitution model and FreeRate heterogeneity, 100 bootstraps, 1,000 replicates of the SH-aLRT branch test and approximate Bayes test. DnaSP software (Rozas et al., 2017) was used to estimate the following variables: number of haplotypes (h), haplotype diversity (Hd), number of polymorphic loci (Ps), nucleotidic diversity (Pi) and Fst values. With the use of DnaSP .nex archives (nexus format) median joining haplotype networks were generated in PopART (Leigh & Bryant, 2015).

A map showing the distribution of the most abundant 16S rRNA-COI haplotypes in the different locations was generated using ggplot2 (v. 3.4.2) (Wickham, 2016) and scatterpie (v.0.2.1) (Yu, 2023). Populations were defined based on the Spalding et al. (2007) ecoregions, except in the case of the Aegean Sea where the ecoregion was divided into North and South Aegean Sea. FST values between populations were calculated using the adegenet (v.2.1.7) (Jombart, 2008), pegas (v.1.2) (Paradis, 2010) and hierfstat packages (v.0.5.11) (Goudet & Jombart, 2022). The effect of the population on genetic differentiation was tested with the function test.g of the hierfstat package. AMOVA (Analysis of Molecular Variance) was performed to determine genetic variation between populations using poppr (v.2.9.4) (Kamvar, Tabima & Grünwald, 2014) and pegas packages. Isolation by distance was tested using a Mantel test between a matrix of genetic distances (calculated using Edwards’ distance) and a matrix of geographic distance between populations (calculated using Euclidean geographic distances) using adegenet and MASS (v.7.3.57) (Venables & Ripley, 2002) packages. Discriminant analysis of principal components (DAPC) and principal component analysis (PCA) were performed with the adegenet package. hierBAPS was also performed as a method for hierarchical clustering of the sequence data to reveal nested population structure, with the use of rhierbaps (v.1.1.4) (Cheng et al., 2013), phytools (v.1.5.1) (Revell, 2012) and ggtree (v.3.2.1) (Yu et al., 2017) packages. Plots were created using ggplot2. All the aforementioned analyses were performed in R version 4.1.1 (R Core Team, 2021).

The map of the sampling sites was generated with the QGIS software. Raw DNA sequences from the present study are available from the European Nucleotide Archive (ENA) (Burgin et al., 2023) at http://www.ebi.ac.uk/ena/data/view/OX406989-OX407068 (16S rRNA) and http://www.ebi.ac.uk/ena/data/view/OX407172-OX407248 (COI).

Results

When examining the shell’s dimensions, the relationship between the height above sediment with the height buried inside the sediment was quite linear for all the study areas (Fig. S3A; Table S3). However, in the regression between the total height and the width, the population from Lesvos followed a different pattern (Fig. S3B; Table S3). In addition, when the P. nobilis individuals are classified into age classes based on the total shell’s height (Butler, Vicente & de Gaulejac, 1993; Richardson et al., 1999; Tempesta, Ciriaco & Del Piero, 2013), it is evident that juveniles were only present in Karpathos and Amvrakikos, while the majority of the individuals were adult juveniles (Fig. S4).

Figure 2 Haplotype network for the 16S rRNA-COI dataset of Greece (Eastern Mediterranean).

Circle size depicts the haplotype frequency; color coding according to sample location; details on the number of samples, sequence size and number of haplotypes are available in Table 2.

Overall, 36 (out of 60) amplifications of eDNA samples were successful for the 16S rRNA and 33 (out of 60) for the COI gene. All the tissue samples provided successful amplifications for both genes. The first dataset included 100 sequences (N) (concatenation of COI and 16SrRNA genes) of 982 bp from the Greek coastline (Eastern Mediterranean Sea). It revealed 34 haplotypes and 45 polymorphic sites. The haplotypic diversity was high (Hd: 0.91 ± 0.017) while the nucleotide diversity was low (Pi: 0.00304 ± 0.00029) (Table 2). The haplotype network had a star-like shape, with two central haplotypes from which all the other haplotypes derive (Fig. 2). The regions of Epanomi and Aggelochori from the North Aegean Sea along with Chios Island and Korinthiakos Gulf formed a distinct group compared to all the other regions. A similar indication of differentiation appeared also in the South Aegean with samples mainly from Karpathos being distinct from the other ones. AMOVA showed that there is a very small variation (0.52%) between the populations but it is much higher, and statistically significant, between the samples (82.02%) (Table 3). FST values were quite low (Table S4), which also implied that there is no differentiation among the populations. When we tested for isolation by distance with the Mantel test, there was no clear isolation by distance pattern (Observation: −0.9748659; Simulated p-value: (1); however, in the scatterplot between the genetic distances and the geographic distances, two clouds of points appeared indicating that there might be distant patches (Fig. S5A). DAPC, after a-score optimization (10 PCs retained), showed that there are a few admixtured individuals (Fig. S5B) but overall DAPC classification is consistent with the original populations; reassignment to actual population was higher for the South Aegean Sea (>80%), followed by the North Aegean Sea (>60%) and the Ionian Sea (>40%). PCA also showed that the population clusters were not very clear (Fig. S5C). hierBAPS clustered the sequences into two main clades and two reduced ones (cluster log marginal likelihood: −444.275995882745) (Fig. S5D), which did not correspond to the geographic location of the populations. Interestingly, the fourth clade contained only three individuals from Epanomi, with the rest of them being found in clades 1 and 2.

Table 3 AMOVA table using genetic distances based on haplotype frequencies of the P. nobilis populations.

	Scenarios	Source of variation	Degrees of
freedom	Sum of squares	Components of covariance (Sigma)	Variation (%)	P-value	
Greece (Eastern Mediterranean) (16S rRNA-COI)	Ionian Sea (Amvrakikos, Korinthiakos) - North Aegean Sea (Lesvos, Aggelochori, Epanomi, Chios, Vourvourou) - South Aegean Sea (Attica, Diafani, Tristomo, Astakida, Crete)	Between populations	2	18.51356	0.01582342	0.52	0.28	
Between samples within population	9	57.22781	0.53133279	17.46	0.01	
Within samples	88	219.5786	2.49521173	82.02	0.01	
Central, Western and Eastern Mediterranean Sea (16S rRNA-COI)	Adriatic Sea (Venice) - Ionian Sea (Amvrakikos, Korinthiakos, Sicily East) - North Aegean Sea (Lesvos, Aggelochori, Epanomi, Chios, Vourvourou) - South Aegean Sea (Attica, Diafani, Tristomo, Astakida, Crete) - Levantine Sea (Cyprus) - Western Mediterranean (Italy, Sicily West, Corsica, Sardinia)	Between populations	5	213.2511	0.8919529	25.07	0.01	
Between samples within population	13	77.38444	0.2702837	7.60	0.01	
Within samples	275	658.6637	2.3951407	67.33	0.01	
Mediterranean Sea (COI)	Adriatic Sea (Venice) -
Ionian Sea (Amvrakikos, Korinthiakos, Sicily East) -
North Aegean Sea (Lesvos, Aggelochori, Epanomi, Chios, Vourvourou) -
South Aegean Sea (Attica, Diafani, Tristomo, Astakida, Crete) - Levantine Sea (Cyprus) -
Western Mediterranean (Bizerta Lagoon, Italy, Sicily West, Corsica, Sardinia, France, Spain) - Tunisian Plateau/Gulf of Sidra (El Ketef, Stah Jaber, Kerkennah Island, El Bibane Lagoon)	Between populations	6	32.72616	0.09382413	9.56	0.01	
Between samples within population	19	20.06477	0.01168851	1.19	0.06	
Within samples	423	370.5275	0.87595168	89.26	0.01	

The second dataset included 294 concatenated sequences (N) of 714 bp from the Western and Eastern Mediterranean Sea (COI and 16SrRNA). It revealed the highest number of haplotypes (104) and polymorphic sites (72) among all datasets. The haplotypic diversity (Hd: 0.961 ± 0.005) was high and the nucleotide diversity was moderately high (Pi: 0.00511 ± 0.00019) (Table 2). The haplotype network showed a clear differentiation among the three subregions; Adriatic Sea, Western and Eastern Mediterranean Sea (Ionian, North Aegean, South Aegean, Levantine) (Fig. 3). A few central, highly frequent haplotypes from the Western Mediterranean Sea split into many closely related unique haplotypes in a star-like scheme. The same structure was observed in the haplotypes that occurred in the Eastern Mediterranean although there were a few that were closer to the Western Mediterranean ones. The Venetian lagoon samples (Adriatic Sea), although distinct, showed a higher relatedness to the Western Mediterranean samples than the Eastern ones, thus strengthening the genetic structuring between Western and Eastern Mediterranean. When the 19 most abundant haplotypes were plotted (67% cumulative abundance), it was again evident that there is a population differentiation across the Mediterranean Sea (Fig. 4). AMOVA also showed a high variation between the populations (25.07%), but again a higher one between the samples (67.33%), with both values being statistically significant (Table 3); however, FST values were quite low (Table S5). When we tested for isolation by distance with the Mantel test, there was no clear pattern (Observation: 0.05073212; Simulated p-value: 0.401) and there was one single consistent cloud of point in the scatterplot, without discontinuities indicating patches (Fig. S6A). DAPC, after a-score optimization (12 PCs retained), showed that there are again a few admixtured individuals (Fig. S6B) but overall DAPC classification is consistent with the original populations; reassignment to actual population reached 100% for the South Aegean Sea population and was higher than 40% for all populations, except for the Ionian Sea one (>10%). PCA also did not reveal a clear clustering (Fig. S6C). hierBAPS clustered the sequences into two main clades and a third, more reduced one (cluster log marginal likelihood: −1485.08119596456) (Fig. S6D), which again was not according to geographic location.

Figure 3 Haplotype network for the 16S rRNA-COI dataset of the Central, Western and Eastern Mediterranean Sea.

Circle size depicts the haplotype frequency; color coding according to sample location; details on the number of samples, sequence size and number of haplotypes are available in Table 2.

Figure 4 Map showing the distribution of the most abundant haplotypes of dataset 2 (16S rRNA-COI Central, Western and Eastern Mediterranean Sea).

Credits: Christina Pavloudi. CC0. Map created using R.

The third dataset included 450 sequences (N) of the COI gene (243 bp) again from the whole Mediterranean Sea. It revealed 48 haplotypes and 36 polymorphic sites. The haplotypic diversity (Hd: 0.652 ± 0.024) was lower than the one observed in the other two datasets and the nucleotide diversity was moderately high (Pi: 0.00475 ± 0.00028) (Table 2). AMOVA showed a low, but statistically significant, variation between the populations (9.56%) and a much higher variation between the samples (89.26%), again statistically significant (Table 3). FST values were again quite low (Table S6). Again, there was no clear pattern of isolation by distance (Observation: 0.2545314; Simulated p-value: 0.213) and one single consistent cloud of points in the scatterplot (Fig. S7A). DAPC, after a-score optimization (nine PCs retained), showed a very similar pattern for most of the individuals (Fig. S7B). Reassignment to actual population was very low for the Ionian Sea, North Aegean Sea and Tunisian Plateau/Gulf of Sidra (<10% in all cases); however, it was >40% for the Western Mediterranean and Levantine Sea populations. PCA again did not reveal a clear clustering (Fig. S7C) and hierBAPS clustered the sequences into three clades, as for the second dataset (cluster log marginal likelihood: −1013.22748489699) (Fig. S7D); the third smaller clade, contained sequences from diverging individuals both from the Eastern as well as from the Western Mediterranean.

Discussion

Morphology of pen shells

As it has been reported, growth rates of P. nobilis are variable based on the availability of zooplankton, which is reflected in the location of the individuals (Richardson et al., 1999). In addition, it has been estimated that the average growth rate for juveniles is 0.28 mm per day, i.e., ∼10 cm per year (Hendriks et al., 2012). According to the distribution of total height of individuals in the studied populations, more than half of our samples are categorized as adult juveniles, and should probably be 2–8 years old (Richardson et al., 1999). Lack of juveniles in all populations except of the population in Karpathos and Amvrakikos, may have been related to the infection of the populations and the subsequent MME.

Population genetic structure

This study contributes significantly to the knowledge of P. nobilis genetic structure as it provides data from regions of the Greek coastline that had not been sampled before. Currently, to the best of our knowledge, the only population with living individuals of P. nobilis is the one in Amvrakikos Gulf; all the other sampled locations have no individuals surviving the MME.

The results of this study also indicate that within the Eastern Mediterranean Sea there is no differentiation among the different geographic regions that were sampled implying a high connectivity among them, i.e., the isolation by distance of the populations of North and South Aegean Sea, as well as of Ionian and Aegean Sea is not relevant. Similar results have been found for the horse mussel (Modiolus barbatus), a fact which was attributed to the very long (up to 6 months) pelagic larval stage of the species (Giantsis et al., 2019), which, however, exceeds by far that of P. nobilis. A similar high connectivity trend was also indicated for the Western Mediterranean P. nobilis populations; this could suggest the possible occurrence of ecological/biological traits, which are typical of the species in the whole Mediterranean basin and affect the species’s population structuring. Another potential explanation for the lack of differentiation between the Eastern Mediterranean Sea populations could be transplantations of individuals. Transplantations have been suggested to be responsible for the absence of geographic structure of Mytilus galloprovincialis populations in the Aegean Sea (Giantsis, Kravva & Apostolidis, 2012). This might have been the case also for P. nobilis, as transplantations had been proposed as a conservation action for the protection of the species (Katsanevakis, 2016; Acarli, 2021), although they were most likely performed only on a local scale and, thus, they should not have influenced the genetic structure of P. nobilis at the scale of the Aegean Sea. However, since they were not documented in detail, it is impossible to fully assess their potential effect on the populations. A slight population differentiation is observed between the regions of the North and South Aegean Sea (Fig. 2). For the South Aegean, this could be attributed to the higher number of collected samples compared to the other regions, leading to a higher haplotypic diversity in this case. On the other hand, it could be attributed to the fact that the island of Karpathos is part of a marine protected area (MPA). Although the design of MPAs is generally not based on genetic and genomic data (Sandström et al., 2016; Xuereb et al., 2020), in certain cases it has been shown that they succeed in preserving most of the genetic diversity of their keystone species (Miller & Ayre, 2008), and combined with the protection measures for those species, they might end up preserving a higher number of haplotypes.

In the haplotype network of the Eastern Mediterranean (Fig. 2) the haplotypes of North Aegean (Epanomi, Aggelochori, Chios) formed a subgroup shown in blue coloring; yet the Korinthiakos Gulf (Ionian Sea) also shares them. These haplotypes were described by Katsares et al. (2008) and were grouped with the ones from the Tunisian coasts in the research of Sanna et al. (2013), reinforcing the hypothesis of the high connectivity within the Eastern Mediterranean basin. On the other hand, the populations that were sampled within the present study (sampled in the period 2018–2021) did not share the above mentioned haplotypes. The intervening period between the studies coincided with the outbreak of the MME, thus raising questions on the association of the populations genetic structuring and the mass mortality events the populations of the species underwent. Unfortunately, it is not possible to estimate the potential genetic structuring of the populations if the MME had not occurred, since the majority of the populations have not survived it. The only exception is the population in Amvrakikos Gulf, for which there is no pre-MME genetic information.

The findings of this study support the distinction of the P. nobilis individuals into three regions of the Mediterranean Sea. The case of the Adriatic Sea is explained in detail in Sanna et al. (2013); it is a semi-enclosed sea where the genetic flow from the rest of the Mediterranean Sea is not that high. The other basins of the Mediterranean Sea are distinct for a number of other species (Penaeus (Melicertus) kerathurus: (Zitari-Chatti et al., 2009); Pomatoschistus tortonesei: (Mejri et al., 2009); Holothuria polii: (Gharbi & Said, 2011); Carcinus aestuarii: (Deli, Said & Chatti, 2015), including P. nobilis (Sanna et al., 2013). The present study analyzed a high number of samples from the Eastern Mediterranean Sea in order to confirm this pattern. The concatenation of the COI and 16S rRNA genes that was used in the present study has also proved useful and more informative in other genetic studies of bivalves (Yuan, He & Huang, 2009; Feng et al., 2011; Slynko et al., 2018), and shows that there is a certain level of differentiation between the P. nobilis populations in the Western vs Eastern Basin. Consistently with the Mediterranean pattern of diffusion already proposed in Sanna et al. (2013) for this species, this finding suggests that the already known oceanographic barriers at the Sicily Strait and at the Otranto Strait might be limiting the dispersal of the species and minimizing the gene flow (Čekovská et al., 2020). Due to its short pelagic larval duration stage, P. nobilis is a species which is considered to be rather affected by currents and fronts; at the same time, it could be less prone to gene flow from other locations (Pascual et al., 2017) and it could exhibit strong population structuring, as has been shown for other bivalves also characterized by a short planktonic larval stage (Ye, Wu & Li, 2015).

eDNA and mtDNA marker sequencing

eDNA has been used widely for biodiversity assessments (Pereira et al., 2021) and for the detection of cryptic, threatened (Hunter et al., 2018) and invasive species (Ardura et al., 2015). This study was the first, to our knowledge, to use eDNA collected separately from each individual for genetic variation assessment on a critically endangered species, although its potential has been advocated for in the literature (Barnes & Turner, 2016; Adams et al., 2019). Our results suggest that the approach can be replicated to other organisms where minimal disturbance and non-invasive methods are in order. In addition, it can be employed in the few remaining populations of P. nobilis around the Mediterranean, such as the ones in Ebro Delta (Prado et al., 2020), the Occitan coast (Peyran et al., 2022) and the one in Amvrakikos Gulf. Successful amplification for our chosen markers was possible for about half of the samples, which is lower compared to the amplification success from the tissue samples, as was originally expected. However, this number is still considered adequate for the estimation of population genetics indices. Another advantage of this approach is the certainty that each sample of genetic material corresponds to a specific individual which would not have been possible if the eDNA matrix was e.g., water or sediment collected from the study sites; however, there have been studies on population-level inferences from eDNA water samples mostly regarding large populations of fish (Sigsgaard et al., 2020).

The results of the present study are based on the sequencing of two mtDNA genes and there is the possibility that they would be different if another approach was used instead or in complement to ours, such as sequencing of microsatellites markers (Meenakshi, Remya & Sanil, 2010; Vanhaecke et al., 2012) or ddRAD sequencing (Darschnik et al., 2019; Ortiz et al., 2021) or even the addition of more mtDNA markers (e.g., D-loop) (Pourkazemi, Skibinski & Beardmore, 1999; Parmaksiz, 2019); using more detailed approaches would enhance the assessment of the genetic structure of P. nobilis throughout the Mediterranean. However, as mentioned previously, P. nobilis is a critically endangered species and the number of available samples for deciphering population genetic structure is quite limited; thus, it is challenging to detect the remaining populations of the species and obtain the appropriate number of samples, with a subsequent high DNA quality, while, at the same time, ensuring the well-being of the organisms.

Conclusions and Prospects Ahead

The present study is the first one that includes such a high number of P. nobilis specimens from different areas of the Eastern Mediterranean basin. Therefore, it significantly contributes to the knowledge of the genetic variability of the pen shell’s populations; the number of available COI sequences has increased 3-fold, while the number of 16S rRNA sequences has had a 20-fold increase. In light of the MME, coordinated studies on the genetic diversity of P. nobilis throughout the Mediterranean Sea, with the cooperation of researchers, institutes and universities, should be performed towards the aim of the conservation and management of the remaining populations of the species. An orchestrated attempt of a pan-Mediterranean investigation appears to be indispensable. Scientific cooperation and use of common standards should be implemented in order to obtain more FAIR data and therefore lead more efficiently to knowledge (Wilkinson et al., 2016). In future conservational plans on a national level, the Eastern Mediterranean basin should be considered as homogenous, based on the findings herein. It is obvious that more samples from the Southern-Eastern Mediterranean Sea would shed more light on the population genetics status of the species.

Supplemental Information

Supplemental Information 1 Sampling brushes used for the eDNA sampling

Click here for additional data file.

Supplemental Information 2 Map showing the locations from which sequences of 16S rRNA, or COI, or both were compiled

Click here for additional data file.

Supplemental Information 3 Scatterplot showing the relationship between the shell’s height above sediment with the shell’s height buried inside the sediment for the P. nobilis individuals sampled for this study

(A) Scatterplot showing the relationship between the shell’s height above sediment with the shell’s height buried inside the sediment for the P. nobilis individuals sampled for this study. (B) Scatterplot showing the relationship between the shell’s total height and the shell’s width for the P. nobilis individuals sampled for this study.

Click here for additional data file.

Supplemental Information 4 Scatterplot showing the relationship between the shell’s total height and the shell’s width for the P. nobilis individuals sampled for this study

The shell’s total height distribution categorized in age classes for the different individuals sampled for this study.

Click here for additional data file.

Supplemental Information 5 Isolation by distance scatterplot for the 16S rRNA-COI Greece (Eastern Mediterranean) dataset

(A) Isolation by distance scatterplot for the 16S rRNA-COI Greece (Eastern Mediterranean) dataset. (B) Membership probability of individuals to populations, as calculated by DAPC, for the 16S rRNA-COI Greece (Eastern Mediterranean) dataset. (C) PCA plot depicting the similarity of different populations for the 16S rRNA-COI Greece (Eastern Mediterranean) dataset. (D) Phylogenetic tree of the 16S rRNA-COI Greece (Eastern Mediterranean) dataset showing the hierBAPS clustering into different groups.

Click here for additional data file.

Supplemental Information 6 Membership probability of individuals to populations, as calculated by DAPC, for the 16S rRNA-COI Greece (Eastern Mediterranean) dataset

(A) Isolation by distance scatterplot for the 16S rRNA-COI Central, Western and Eastern Mediterranean Sea dataset. (B) Membership probability of individuals to populations, as calculated by DAPC, for the 16S rRNA-COI Central, Western and Eastern Mediterranean Sea dataset. (C) PCA plot depicting the similarity of different populations for the 16S rRNA-COI Central, Western and Eastern Mediterranean Sea dataset. (D) Phylogenetic tree of the 16S rRNA-COI Central, Western and Eastern Mediterranean Sea dataset showing the hierBAPS clustering into different groups.

Click here for additional data file.

Supplemental Information 7 PCA plot depicting the similarity of different populations for the 16S rRNA-COI Greece (Eastern Mediterranean) dataset

Click here for additional data file.

Supplemental Information 8 Details and metadata of the samples

TH: Total height, HS: Height above sediment, HD: Height inside the sediment, W: Greater width, DNA conc: DNA concentration of the samples (ng/µl); where DNA was extracted for multiple tissues, the average DNA concentration is shown.

Click here for additional data file.

Supplemental Information 9 Accession numbers and location of publicly available sequences used for the analyses

Click here for additional data file.

Supplemental Information 10 The linear regression equations for the different study sites and the shell dimensions of the individuals

Click here for additional data file.

Supplemental Information 11 Pairwise FST values comparing populations in the Ionian, North Aegean and South Aegean

p-value (population effect on genetic differentiation): 0.01.

Click here for additional data file.

Supplemental Information 12 Pairwise FST values comparing populations in the Adriatic Sea, Ionian Sea, North Aegean Sea, South Aegean Sea, Levantine Sea, Western Mediterranean

p-value (population effect on genetic differentiation): 0.01.

Click here for additional data file.

Supplemental Information 13 Pairwise FST values comparing populations in the Adriatic Sea, Ionian Sea, North Aegean Sea, South Aegean Sea, Levantine Sea, Western Mediterranean, Tunisian Plateau/Gulf of Sidra

p-value (population effect on genetic differentiation): 0.01.

Click here for additional data file.

We would like to thank the Management Agency of the Dodecanese Protected Areas (MADPA) and especially Mr Dinos Protopapas and Mr Giorgos Prearis (captain of the R/V Saria) for providing assistance during our sampling campaign in Karpathos Island. The authors would like to also thank Dr Vassilis Gerakaris and Hippocampus Bali dive center for assistance during the sample collection. We would also like to thank Dr Katerina Vasileiadou (ORCID: 0000-0002-5057-6417) and Ms Xenia Sarropoulou (ORCID: 0000-0003-3671-9693) for their invaluable contribution to the manuscript and the statistical analyses.

Additional Information and Declarations

Competing Interests

Author Contributions

Field Study Permissions

Data Availability

The authors declare there are no competing interests.

Georgia Sarafidou performed the experiments, analyzed the data, prepared figures and/or tables, authored or reviewed drafts of the article, and approved the final draft.

Dimitris Tsaparis analyzed the data, authored or reviewed drafts of the article, and approved the final draft.

Yiannis Issaris performed the experiments, authored or reviewed drafts of the article, resources (Provision of samples), Funding acquisition, Project administration, and approved the final draft.

Giorgos Chatzigeorgiou performed the experiments, prepared figures and/or tables, authored or reviewed drafts of the article, resources (Provision of samples), and approved the final draft.

Panos Grigoriou performed the experiments, authored or reviewed drafts of the article, resources (Provision of samples), and approved the final draft.

Eva Chatzinikolaou conceived and designed the experiments, authored or reviewed drafts of the article, resources (Provision of samples), and approved the final draft.

Christina Pavloudi conceived and designed the experiments, performed the experiments, analyzed the data, prepared figures and/or tables, authored or reviewed drafts of the article, funding acquisition, Project administration, and approved the final draft.

The following information was supplied relating to field study approvals (i.e., approving body and any reference numbers):

Samples were collected under a research permit (175828/2195 of 14/11/2018) issued by the Greek Ministry of Environment and Energy, General Directorate for the Forests & Forest’s Environment, Department of Wildlife and Hunting Management.

The following information was supplied regarding data availability:

The raw sequences are available at the European Nucleotide Archive (ENA): 16S rRNA, OX406989-OX407068 and COI, OX407172-OX407248.

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
