# Peer review of "Insights on Pinna nobilis population genetic structure in the Aegean and Ionian Sea"

_PeerJ, doi:10.7717/peerj.16491_

## Round 0.1 · original submission · Major Revisions

Both reviewers appreciate the work you've put into this paper and have provided a number of comments that I think will improve the manuscript. Reviewer 1 has included an annotated version of the manuscript, with detailed comments.

·

Basic reporting

General Comment:
I have reviewed the manuscript titled “Insights on Pinna nobilis genetic connectivity in the Eastern Mediterranean Sea” by Sarafidou et al.
In this study, the authors evaluated the genetic structure of Pinna nobilis, a critically endangered species, based on mtDNA and 16S rDNA sequences. This work includes samples from Greece that were newly collected and sequences previously published in GenBank.
The authors also tested a non-invasive method, which is of interest and importance in the context of a highly endangered species. However, even if this method is valuable, it is not comparable to eDNA methods used for the detection of cryptic or invasive species, as suggested by the authors.
It is a valuable study, as it brings more knowledge about the genetic structure of populations in Greece, which was poorly documented before, in a context where fan mussels are suffering a pandemic that is devastating all populations throughout the Mediterranean Sea. This study will greatly contribute to the conservation of Pinna nobilis. However, I believe that data analysis needs more attention and detail. Some information about sampling design is missing to interpret results and some basic genetic analysis could be implemented to improve the article such as FST, genetic distance, a mantel test, etc. The structure of the manuscript and the presentation of the results are not very clear to me, and some points need clarifying as authors seem to do groups of populations that are not explained. I would rather focus on a deeper analysis of the samples they collected and the genetic structure of the areas and compare with what is already known in these areas, knowing that a genetic study was previously by Katsares et al. 2008 (DOI: 10.2478/s11756-008-0061-8). Is there a change in genetic diversity, in the structure? Then do a more global analysis by adding sequences available in GenBank.
Also, this study lacks a statement of the actual situation of the fan mussel population in the study area. The study spans from 2018 to 2021 which is the period during which all sea populations were devastated throughout the Mediterranean Sea. The existence of living and healthy Pinna nobilis populations in the sea is very important information for conservation purposes.
Please, see my detailed comments in the attached PDF.

Experimental design

no comment

Validity of the findings

no comment

Additional comments

no comment

Reviewer 2 ·

Basic reporting

I found the content of this manuscript relevant and highly interesting for researchers that work in this field of research. The genetic variation of Pinna nobilis from Eastern Mediterranean is poorly known and for this reason a relevant increasing of knowledge represents a tool to better manage the crisis that this species is experiencing.
Even if I’m not a native English speaker, I found the text well-written and easy to be understood in most of its parts.
However, according to my opinion the manuscript needs some relevant improvements before a possible acceptance.

Experimental design

My main concerns are below reported for each section:
Introduction: It is well written and addressed. The references are properly included.
Material and Methods: In my own opinion both experimental and sampling plans are somewhat confused and need to be better explained according to the following suggestions.
I found the eDNA sampling highly useful and deserving to be better tested. However, I was wondering if you did test the goodness of Pinna nobilis DNA extraction obtained by eDNA sampling. I mean, for the individuals whose tissue was used for the DNA extraction, did you try also to collect a sample of eDNA in order to verify if the mitochondrial sequences obtained from eDNA belong to the individual and perfectly match the ones obtained from DNA extraction from tissue?
For what concerns the COI primers that you firstly used for PCR, as you tested the primers provided by Folmer et al., 1994, Sanna et al., 2013, 2014, and Leray et al., 2013, why you did not test the primers provided by Katsares et al. 2008 for the Aegean populations?
For what concerns the analyses which were performed, please deposit your sequences also on GenBank in order to make them easy to be used from a group of researchers as large as possible.
Furthermore, in my own opinion the analyses performed are a little outdated and Authors should also insert a clustering analysis to support AMOVA that could be moved to supplementary material in the revised version of the manuscript. They should perform a PCoA plus a Bayesian clustering analysis such as BAPS.
Results: The criteria used by the Authors to create 8 different datasets are somewhat confused for the reader. Please reduce the number of datasets in order to simplify the analyses and phylogeographic results. The datasets that must be analysed could be 3: whole Mediterranean, central-Western Mediterranean and Eastern Mediterranean. The fine discrepancies among macro-populations within datasets will be evidenced by clustering analyses without need to analysed a too large number of datasets.
Furthermore, why you did not use the COI sequences from Spain in Western Mediterranean provided by Wesselmann et al. 2018? These sequences should be included in the dataset which is called “whole Mediterranean”. Furthermore again, a table clearly reporting which populations are included in the analysed dataset is necessary. I did not fully understand when Tunisian sequences from Raboui et al. 2011 are included in a dataset and in which dataset the sequences from Cyprus from GenBank were included by Authors.
The main concern is that this whole part of the manuscript, in my opinion should be re-written and better organized in order to simplify the text and fully clarify the genetic structuring, if any, among areas.

Validity of the findings

no comment

Additional comments

no comment

---

## Round 0.2 · Major Revisions

Thanks for the work on the revision; the paper has been seen by two reviewers and they have included a number of comments that will still need to be addressed prior to publication. In particular, please see the comments from Reviewer 1 about tables and figures.

·

Basic reporting

General Comment:
I have thoroughly reviewed the revised version of the manuscript titled "Insights on Pinna nobilis Genetic Structure in Greece" by Sarafidou et al.
In this study, the authors have explored the genetic structure of Pinna nobilis using mtDNA and 16S rDNA sequences. Their work encompasses newly collected samples from Greece and is completed with sequences previously published in GenBank.

I appreciate the authors' dedication to addressing my previous comments and incorporating them into the manuscript. They have made substantial improvements by clarifying numerous aspects and introducing new interesting analyses.

I continue to regard this study as highly valuable, contributing crucial insights that will be useful for the conservation of the species. This is particularly pertinent in light of the ongoing pandemic devastating fan mussel populations throughout the Mediterranean Sea.

However, I still believe there is room for further enhancement and clarification of the manuscript. Specifically, some figures and tables, including those in the supplementary materials, could benefit from improvement. Additionally, the FST tables lack levels of significance, which would enhance their interpretability. To enhance readability and comprehension, it would be beneficial to consider subdividing the results section into more clearly defined sections. Another suggestion is to merge certain figures for improved result interpretation, such as Supp Fig. 6 + 7 or Supp Fig 10 + 11, etc.

I still think it would be preferable to remove data related to height and width, as it is not directly involved in analyses presented in the paper. While I understand the authors' intent to provide comprehensive metadata and value their work, this information seems somewhat drowned in a sea of data. Given the abundance of figures and tables in the supplementary materials (16 supp figs and 9 supp tables), its inclusion may add unnecessary complexity. Maybe authors could consider writing a separate article discussing variations in individual size and the factors contributing to morphological differences.

Also, regarding the term 'eDNA sampling,' I understand why the authors consider this sampling as eDNA since it involves sampling mucus rather than tissues directly. I do believe that the description of this technique is very valuable and of great interest. However, I have some doubts about whether this terminology is appropriate in this context. eDNA typically aims to describe the genetic material present in an environment. Here, the approach is different because it involves non-invasive sampling of an individual that is clearly identified. In this case, it seems to me that the issues encountered are rather different. An eDNA approach would have been to sample water and identify the number of haplotypes found in the environment, then compare this estimation to the estimation obtained through conventional population sampling. However, if the editor and the other reviewer agree with this use of the term eDNA, then okay.
Please find more detailed comments in the attached PDF.

Experimental design

no comment

Validity of the findings

no comment

Additional comments

no comment

Reviewer 2 ·

Basic reporting

I truly appreciate the new version of the manuscript. Authors strongly improved it following the suggestions of the Reviewers. The English form is clear and well-written, only a few parts could be further improved during the last round of revision. The contents of this study are interesting and deserving to be published after just a few minor revisions. The references are properly chosen and perfectly describe the background in which this study would be included after its pubblication.
I inserted my changings (in BLUE font), my comments and my suggestions directly in the marked-up word file that was provided by the Authors. You will find it as a PDF attachment.

Experimental design

The aims of the study deserve to be investigated and the Authors used the proper bioinformatic tools to infer on the genetic variation of Pinna nobilis from sites that have been never investigated before.
The methods are well addressed, I appreciate the chosing of some analyses to reach the goals of the study.

Validity of the findings

Results are well-descripted. In my opinion the manuscript is a very good job and provides new interesting molecular data that can be useful for the conservation (if possible) of Pinna nobilis.

Annotated reviews are not available for download in order to protect the identity of reviewers who chose to remain anonymous.

---

## Round 0.3 · accepted · Accept

Thanks for addressing the reviewers' previous comments in your revision -- you've done a good job responding, and the resubmission is improved.